# Effects of Ethylene-Propylene-Diene Monomers (EPDMs) with Different Moony Viscosity on Crystallization Behavior, Structure, and Mechanical Properties of Thermoplastic Vulcanizates (TPVs)

**DOI:** 10.3390/polym15030642

**Published:** 2023-01-26

**Authors:** Li-Fu Song, Nan Bai, Ying Shi, Yuan-Xia Wang, Li-Xin Song, Li-Zhi Liu

**Affiliations:** 1Advanced Manufacturing Institute of Polymer Industry, Shenyang University of Chemical Technology, Shenyang 110027, China; 2College of Materials Science and Engineering, Shenyang University of Chemical Technology, Shenyang 110027, China

**Keywords:** EPDMs with different Moony viscosity, crystallization behavior, compatibility, structure, mechanical properties

## Abstract

Moony viscosity of ethylene-propylene-diene monomers (EPDMs) can have effect on the crystallization dynamics, structure, and properties of EPDM/polypropylene (PP)-based thermoplastic vulcanizates (TPVs). TPVs with two different Moony viscosities are prepared via a twin-screw extruder, respectively. Crosslinked EPDM with lower Moony viscosity has a higher crosslinking density and the nucleation effect of its crosslink point improves the crystallization ability of PP in TPV, leading to PP phase crystallization at higher temperatures. For TPV with an EPDM of higher Moony viscosity, it has higher crystallinity and the EPDM phase crystallized earlier. Synchrotron radiation studies show that the EPDM with low Moony viscosity has no obvious crystalline structure, and the prepared TPV has an obvious phase separation structure, while the TPV with higher Mooney viscosity of the EPDM does not exhibit obvious phase separation, indicating that the longer EPDM chains have better compatibility with PP in TPV, also evidenced by the almost disappearance of the PP glass transition peak in TPV, from the dynamic mechanical analysis. The longer EPDM chains in TPV provide more physical entanglement and better interaction with PP molecules, resulting in a stronger strain hardening process, longer elongation at break, and higher tensile stress in TPV.

## 1. Introduction

Ethylene-propylene-diene monomer (EPDM)/polypropylene (PP) thermoplastic vulcanizates (TPVs), with a high proportion (50–80 wt%) of EPDM dispersed in a low content of continuous thermoplastic PP phase [1], are usually prepared by pre-blending elastomer and semicrystalline thermoplastics, followed by dynamic vulcanization. Dynamic vulcanization involves crosslinking high-content elastomer components via an in-situ process and dispersing these in the low-content section of the continuous thermoplastic PP by means of a shearing action [2,3,4,5,6]. The dispersed crosslinked elastomer phase improves the toughness and elasticity of the TPV, whereas the thermoplastic PP phase provides good melt processing properties, and the crystalline structure of the plastic phase also gives the TPV its physical and mechanical properties [7,8,9,10,11,12,13]. Moreover, semicrystalline PP provides relatively high mechanical properties, which can be easily processed and recycled [14]. Meanwhile, EPDMs can also have a similar effect on this property.

Recent studies on EPDM/PP TPVs have focused on phase transition, the crosslinking density of the rubber phase, and the dynamic vulcanization mechanism [4,15,16,17,18,19,20,21,22,23]. Min Shi et al. [4] investigated the effect of different vulcanizing agents (i.e., dicumyl peroxide (DCP)) contents on the morphology and crystallization of TPVs. Avalos BF et al. [20] investigated the effect of the ethylene content of EPDMs and found that TPVs prepared using higher-ethylene-content EPDMs had better mechanical properties. Hanguang Wu et al. [21] explored the real microstructural formation mechanism in TPVs during dynamic vulcanization and found that the rubber particles in TPVs are agglomerated, having a diameter of ~40–60 nm. The crystallization behavior of TPVs has also been extensively studied, but most studies about the crystallization behavior of TPVs primarily focused on the crystallization behavior of PP. However, EPDMs with good chain regularity maintain their crystalline property even after crosslinking. How the crystalline behavior of EPDMs affects the structure and property of TPVs remains a question. Some fundamental issues related to the crystalline behavior of TPVs are important but lack intensive studies, such as studies on the effect of different structures of EPDMs with the same ethylene content on the crystalline behavior, microstructure, and mechanical properties of TPVs.

Here, two EPDM samples with the same ethylene content and different Mooney viscosities were chosen to prepare TPVs with PP in a two-step process. The crystalline structure was studied using Synchrotron wide-angle x-ray diffraction (WAXD). Non-isothermal crystallization dynamics and thermal properties were studied with differential scanning calorimetry (DSC). The phase structures were studied with small-angle X-ray scattering (SAXS). The compatibility between EPDMs and PP was assessed via SAXS and dynamic mechanical properties. The structure–property relationship is also discussed.

## 2. Part of Experiment

### 2.1. Materials and Sample Preparation

The PP (code T30s) with a melt flow index value of 3.0 g/10 min (200 °C, 5 kg) was supplied by Sinopec Shanghai (Shanghai, China). The selected ethylene-propylene-diene monomer (EPDM) selected in this study were granular material with NORDEL™ IP 4770P and NORDEL™ IP 4725P number by DU Pont (DOW, Wilmington, DE, USA); the physical information is listed in Table 1. They are both 70% ethylene and 5% ethylidene norbornene (ENB). The difference between them is the Mooney viscosity: one has a Mooney viscosity of 70 MU (NORDEL™ IP 4770P, ML_1+4_, 125 °C) and the other one is 25MU (NORDEL™ IP 4770P, ML_1+4_, 125 °C).

Dicumyl peroxide (DCP) with 40% active peroxide content (Perkadox-BC-40B-PD) with a half-life (t_1/2_) of 1 h at 138 °C, and the specific gravity at 23 °C is 1.53 g/cm^3^, was obtained from AkzoNobel Chemical Co., Ltd. (Ningbo, China). Additionally, stabilizers used were Antioxidant 1010 (Shanghai Bonus Chemical Co., Ltd., Shanghai, China).

The PP/EPDM TPV: The TPV was prepared via a two-step method. First, an EPDM matrix and DCP were premixed and mixed uniformly on a two-rolling mill at 25 °C. After 5 min of mixing time, the EPDM pre-blends were removed from the mixer. Then, the PP and antioxidant 1010 were added to a micro twin-screw extruder (WLG10, screw small head diameter 7.4 mm, effective length 154 mm, Shanghai Xinshuo Precision Machinery Co, Ltd., Shanghai, China) for self-circulation at 180 °C, 80 r/min. After 3 min, the rotation speed was adjusted to 140 r/min, and the pre-blends of EPDM were added to PP according to the ratio, the self-circulation lasted for 5 min, and the compound was cured during compounding. Crosslinked EPDM was directly prepared from the above pre-blends via pressing at 180 °C and 10 MPa for 5 min on a flat plate vulcanizer (Yf-8017, Yangzhou Yuanfeng Test Machinery Factory, Yangzhou, China). The composition proportion of the neat PP, crosslinked EPDMs, and EPDM/PP thermoplastic vulcanizates are shown in Table 2. The 2 wt% of DCP was selected to crosslink with EPDM.

### 2.2. Thermal Analysis

Thermal behaviors of all the samples were studied via a DSC (Q100 system, TA, New Castle, DE, USA) equipped with the liquid nitrogen cooling accessory. The test sample was sealed in an aluminum holder and heated with nitrogen at a rate of 10 °C /min from room temperature to 200 °C, which is used as an environment and replaced oxygen in the system. Then, the samples were cooled to −20 °C at a rate of 10 °C /min to study their non-isothermal crystallization. The samples were subsequently heated back to 200 °C at the same rate in order to investigate their melting behavior.

### 2.3. Synchrotron SAXS/WAXD Measurements

SAXS and WAXD measurements were performed at the synchrotron beamline 1W2A of the Synchrotron Radiation Facility, Beijing, China. The energy of the X-ray radiation was 8.052 keV, resulting in a wavelength of 0.1542 nm. The specimen (0.1 × 1 × 3 cm^3^ dimensions) was mounted on a portable background plate on the beam line. During WAXD measurement, Mar165-CCD was set at 96.25 mm sample-detector distance in the direction of the beam for WAXD data collections. The collected data were corrected for air background prior to any analysis.

During SAXS measurement, the beam-stop size was selected as 4 mm, and the distance between the sample and the detector was 1468.06 mm. At this distance, the effective scattering vector q is defined as q = 4πsinθ/λ, where λ is the X-ray wavelength, and θ is half of the scattering angle (2θ). The SAXS patterns were recorded with a 165 mm diameter two-dimensional Mar CCD detector. The collected data were corrected for air background before any analysis. Data processing was performed with the computer program “XPolar”. The SAXS and WAXD patterns were normalized to the primary beam intensity and corrected for background scattering.

### 2.4. Mechanical Testing

The sample was prepared using a cutting machine for dumbbell-shaped sample bars with dimensions of 75 mm length, 1 mm thickness, 25 mm neck length, and 4 mm neck width. The deformation of the sample was performed with a tensile apparatus (Autograph AGS-X-20kN, SHIMADZU, Kyoto, Japan). The original length L_0_ between the Shimadzu jaws was 20 mm. Stretching is performed in a symmetric mode where the detection point on the sample remains fixed in the stretching space. The experiments were carried out at room temperature (25 °C). During the deformation process, the tensile rate of the specimen is constant (100 mm/min).

### 2.5. Dynamic Mechanical Analysis (DMA)

Dynamic mechanical analysis was carried out in a Diamond DMA machine (Waltham, MA, USA). Specimens were tested in the dynamic tensile mode with a frequency of 1 Hz from −70 °C to 120 °C with a heating rate of 3 °C·min^−1^. Sample dimensions were 10 × 50 × 1 mm.

## 3. Results and Discussion

### 3.1. Crosslinking of EPDMs

T_90_, defined as the optimum cure time of rubber, was measured using a rotorless vulcanizer (MRC3). The crosslinking curves of the EPDM−4770P and EPDM−4725P are illustrated in Figure 1. The Mooney viscosities of the EPDM−4770P and EPDM−4725P are 70 and 25, respectively. As shown in Figure 1, the T_90_ of the EPDM−4725P and EPDM−4770P is 4.4 min and 4.2 min, respectively, exhibiting a little difference. The EPDM−4725P has a much smaller Mooney viscosity than the EPDM−4770P. However, the EPDM−4725P exhibits a significantly higher torque, indicating that crosslinked 4725P has a higher crosslinking density. The higher crosslinking density of the EPDM−4725 with low Mooney viscosity implies that the low Mooney viscosity is probably because of its shorter molecular chains.

### 3.2. Structure of the Crosslinked EPDMs and EPDM/PP TPVs

EPDMs with high ethylene content (≥64–65 wt%) exhibited semicrystalline characteristics at room temperature [4]. Both the EPDM NORDEL™ 4770P and 4725P used in this study have an ethylene content of 70 wt%. β-PP crystals in TPV can have an effect on both mechanical strength and elasticity [3,23]. Synchrotron WAXD was used to study the crystal structure of EPDMs and different EPDM-based TPVs at 25 °C (see Figure 2). In Figure 2a, the characteristic diffraction peak of the EPDM-4770P and crosslinked 4770P appears at 20.83°, which corresponds to the (110) crystal plane of the ethylene crystal, indicating that the crystalline ethylene structure exists in both the uncrosslinked EPDM−4770P and crosslinked EPDM−4770P. The EPDM−4725P just exhibits a diffraction peak at 19.76°, which corresponds to an amorphous region, indicating that no crystals exist at 25 °C.

In Figure 2b, neat PP shows a small diffraction peak at 16.1°, which corresponds to the (300) crystal planes of β crystals in PP [24,25,26]. Changes in the crystalline structure of TPVs prepared via PP and two different EPDMs are shown in Figure 2b. The diffraction peaks of the two TPVs at 16.1° that correspond to the β (300) plane of PP disappeared. This result can be interpreted as follows. The mass content of the EPDM matrix in the TPV is ~60 wt%, and more EPDM molecules are dispersed in the PP matrix and enter the PP phase. The movement of the PP molecular chains is prevented, and the crystallization of β crystals is limited. Besides, the strong shear forces during processing can also inhibit the production of β crystals [24]. The EPDM phase in the TPV also affects the variation of the PP diffraction peaks, and part of the diffraction peaks overlaps with that of the EPDM peaks. The peak at 23° in the 4770P TPV is the (200) crystalline surface of EP segments in EPDMs, which is not found in the 4725P TPV. The scattering peak at 18.80° of the 4770P TPV corresponds to the (110) crystalline plane overlapping with the (111) crystalline plane diffraction peak of PP at 21.4°, resulting in a significant increase in peak intensity.

### 3.3. Non-isothermal Crystallization of the EPDMs, Crosslinked EPDMs, and TPVs, with DSC

For EPDM-based TPVs, because of the presence of numerous propenyl structural units, EPDMs and PP are structurally similar and have some degree of compatibility [27]. The non-isothermal crystallization and melting behaviors of the EPDMs, crosslinked EPDMs, and TPVs were investigated with DSC. The cooling and second heating trace of the EPDMs and crosslinked EPDMs are shown in Figure 3.

Figure 3a shows that both the EPDM−4725P and EPDM−4770P exhibit crystalline peaks at about 22 °C and 58 °C, respectively, corresponding to crystallization of the ethylene chains in the EPDMs, whereas the EPDM−4770P shows a higher main crystallization peak and a better peak shape than the EPDM−4725P. This outcome can be attributed to longer ethylene chains in the EPDM−4770P, resulting in the crosslinked 4770P having better crystallization ability than 4725P. The EPDM-4770P still exhibits an obvious crystalline peak after crosslinking via 2 phr of DCP. This result reveals that crosslinked 4770P has more ordered crystallizable molecular chain segments and longer molecular chains, which is consistent with the crosslinking curve analysis. Figure 3b shows that the crosslinked EPDMs exhibit lower melting points compared with the neat EPDMs, indicating that the crystals of the crosslinked EPDMs are smaller than those of the neat EPDMs.

The effect of the crosslinked EPDMs on the crystallization behavior of the TPVs is shown in Figure 4. Figure 4a shows that the TPVs show a strong crystallization peak at a temperature of 118–119 °C and a weak crystallization peak at a temperature of <60 °C, corresponding to the crystallization peak of propenyl chains from PP and short ethylene chains from EPDMs. The weak crystallization peak of the 4770P TPV is located at the higher end (60 °C) compared with that of the neat EPDM−4770P (54 °C), whereas the weak crystallization peak of the 4725P TPV is located at the lower end compared with that of the neat EPDM−4725P. This result can be interpreted as follows. In TPVs, the propenyl chain segments of the EPDM phase are compatible with the propenyl chain segments in the PP phase, permitting the propenyl chain segments of EPDMs easy entry into the PP phase. Therefore, the ethylene segment connected with the propenyl chain segments can also be easily moved into the PP phase and is thus affected by PP crystallization, leading to the crystallization of the long ethylene segments of EPDMs in the 4770P TPV at a higher temperature than those of the neat EPDM−4770P. However, the crystallization temperature of ethylene segments in the 4725P TPV is nearly the same as that of the ethylene segments in the crosslinked EPDM−4725P, which is presumably because of the crystallization restriction due to the greater number of crosslinking points and shorter chains of ethylene segments of EPDMs in the 4725P TPV. A higher main sharp crystallization peak temperature (119 °C) is observed for the 4725P TPV than for neat PP (118 °C), resulting from the higher number of crosslinking points in the 4725P TPV that play a nucleation role [4,28,29]. The lower sharp crystallization peak temperature (118 °C) of the 4770P TPV suggests that the mobility of PP molecular chains was hindered by EPDM molecular chains, resulting in a suppressed crystallization ability of PP. Neat PP showed a perfect crystallization peak at 118 °C. After dynamic crosslinking, the crystallization behaviors of PP in TPVs changed slightly. The crystallization peak temperature of PP in the 4770P TPV (118 °C) is the same as that of neat PP, whereas the crystallization peak temperature of PP in the 4725P TPV (119 °C) was 1 °C higher than that for neat PP (118 °C).

Figure 4b shows the secondary heating traces of neat PP, the crosslinked EPDMs, and the EPDM/PP TPV. Crystallinity, evaluated based on the theoretical heat of fusion ∆H_0_ m for 100% crystallized polyethylene being 293 J/g [28] and the theoretical heat of fusion ∆H_0_ m for 100% crystallized polypropylene being 209 J/g [10], is presented in Table 3 and 4. As shown in Table 3, the crystallinity of crosslinked EPDM-4770 is much higher than that of crosslinked EPDM-4725, indicating that the crosslinked EPDM with higher molecular weight has the higher crystallinity. The melting temperatures (T_m_) of both TPVs are lower than that of neat PP (see Table 4). The lower Tm indicates that smaller crystals were formed. Moreover, the melt peak temperatures of the crosslinked EPDMs in the TPVs were similar to those of the neat EPDMs; however, the main melt peak temperature of the 4725P TPV (159 °C) was higher than that of the 4770P TPV (158 °C) because of the stronger nucleation effect of crosslinking points in the 4725P TPV. The 4770 TPV exhibited a higher crystallinity of the PP phase than the 4725 TPV, which resulted from the good compatibility of PP and EPDMs with fewer crosslinks, which was helpful for crystallization.

The above discussion shows that the crosslinking point plays a nucleation role for the EPDM with low molecular weight in the TPV and improves the crystallization ability of PP to some extent, thus resulting in better crystallization than the EPDM with high molecular weight. The compatibility of high molecular weight EPDM with PP was better, which was evidenced by a higher crystallization temperature of the EPDM phase in the TPV. A higher crystallinity of the PP phase was observed for the TPV with high molecular weight EPDM than for the TPV with low molecular weight EPDM, suggesting that the good compatibility of PP and EPDMs with fewer crosslinks is helpful for crystallization, leading to higher crystallization.

### 3.4. Packing Structure of the EPDMs and TPV

The properties of polymers and their blends are closely related to the connection and distribution between the crystallization and amorphous regions [15,30]. SAXS was performed to study the structure of the EPDMs and TPVs at the nanoscale.

A density difference in the materials produced a scattering signal, and the samples studied in the present study had equal thickness. As shown in Figure 5a, crosslinked 4725P exhibited a peak around q at 0.41 nm^−1^, whereas the crosslinked EPDM−4725P did not exhibit the crystallization structure, as discussed in the WAXD result section, indicating that the crosslinked EPDM−4725P had poorly organized chains at room temperature, and the crystals were weak. The EPDM−4770P showed a peak at a higher q value, indicating that the long period of the EPDM−4770P was smaller than that of the EPDM−4725P because of the predominance of amorphous forms in the EPDM−4725P and the larger distance between the phase regions formed by amorphous forms and fewer crystals. After crosslinking, the peak positions of the EPDM−4770P and EPDM−4725P showed different patterns of change. The peak position of crosslinked 4770P remained unchanged, whereas the peak of crosslinked 4725P shifted to a higher q-value end, indicating that crosslinking slightly affected the crystallization region of crosslinked 4770P. The rightward shift of the 4725P peak position indicates that the averaged inter-distance of neighboring crystals becomes smaller, and the weak crystallization becomes more fragmented, implying that crosslinking considerably affects the regularity of the 4725P molecular chains.

Neat PP exhibited a main well-defined scattering peak at q 0.47 nm^−1^, as shown in Figure 5b. For TPVs, the density of PP in the dispersed phase was 0.91 g/cm^3^, and that of the crosslinked EPDM was about 0.86 g/cm^3^, which led to a density contrast between them. The 4725P TPV showed a weak scattering plateau at the low q value (region I), which was because of phase separation between the crosslinked EPDM and PP. In contrast, the 4770P TPV did not show a weak scattering peak in region I, indicating that phase separation between the EPDM and PP was not obvious. Both the TPVs showed only a main well-defined scattering peak in region II, which shifted to a higher peak position than that of the crosslinked EPDM, indicating that an amount of EPDMs entered the phase region of PP and resulted in a small difference in density.

### 3.5. Dynamic Mechanical Analysis

The relationship between the dynamic mechanical properties and temperature of the materials was investigated via dynamic mechanical properties [31,32,33]. The dynamic mechanical properties of neat PP, TPVs, and the crosslinked EPDMs prepared using different EPDMs are shown in Figure 6.

As shown in Figure 6a, crosslinked 4770P exhibited a higher storage modulus (E’) than crosslinked 4725P, resulting from the high crystallinity of crosslinked 4770P and the longer molecular chains. Although the EPDM−4770P had a higher molecular weight than the EPDM−4725P, the storage modulus of the 4725P TPV was comparable to that of the 4770P TPV, indicating that the high crosslinking density of the EPDM−4725P compensated for the low strength of the EPDM−4725P.

The relationship between the loss factor (tan δ) and the temperature of the materials is presented in Figure 6b. Neat PP showed a glass transition temperature (Tg) at 4.6 °C. Crosslinked 4725P showed a narrow glass transition peak at −40 °C. Compared with cross-linked 4725P, crosslinked 4770P showed a broader transition peak because the longer crystalline molecular chains in crosslinked 4770P restricted the movement of the longer amorphous chain segments.

The 4725P TPV exhibited two peaks as follows: a sharp glass transition peak at −39.7 °C, which corresponded to the 4725P rubber phase, a weak glass transition peak at 3 °C (second peak), which was related to Tg of PP [34,35]. This result suggests that PP and rubber exhibit obvious phase separation. However, for the 4770P TPV, the second peak from the amorphous phase of PP nearly disappeared because a part of the EPDM and PP molecules entangled together, indicating the better compatibility of PP with the EPDM−4770P. The 4770P TPV showed a lower tan δ value for the glass transition peak at −39.7 °C than the 4725P TPV, indicating that a lower degree of energy dissipation and fewer amorphous molecular chains generated internal friction.

The above result shows that the PP and crosslinked EPDM with higher molecular weight showed good compatibility in the TPV as evidenced by the glass transition peak in the amorphous phase of PP in the 4770P TPV, which almost disappeared compared with the TPV with molecular weight EPDM.

### 3.6. Different Tensile Properties of the TPVs and the Crosslinked EPDMs

Figure 7 shows the stress-strain curves of the TPVs and the crosslinked EPDMs during uniaxial deformation. Crosslinked 4770P exhibited a stronger strain-hardening process, longer elongation at break, and higher tensile stress than crosslinked 4725P. Despite the higher crosslinking density of crosslinked 4725P than crosslinked 4770P, the mechanical strength of crosslinked 4770P was significantly greater than crosslinked 4725P because of the more physical entanglements of longer molecules and the strain-hardening process promotion by more crystals, which resulted in better mechanical properties.

For TPVs, the network structure formed by the crosslinked EPDM allowed the material to withstand higher stresses. The elongation at the break of the 4770P TPV was significantly larger than that of the 4725P TPV by 106%. Although the crosslinked rubbers were in the dispersed phase, the rubber with longer molecules showed a better interaction with PP molecules. Table 5 shows the tensile strength of the 4770P TPV was 33.6 MPa, which was much higher than that (20.9 MPa) for the 4725P TPV. This result can be interpreted as follows. The tensile strength was related to the intrinsic property of the continuous PP phase and the effect of the dispersed rubber phase. The higher crystallinity of PP in the 4470 TPV and the better compatibility offered higher strength, whereas crystals in the EPDM phase played a reinforcing role, leading to improved mechanical properties.

## 4. Conclusions

The effects of EPDM with different Mooney viscosity on the non-isothermal crystallization, dynamic mechanical property, and mechanical properties of TPV are studied in this study. The crosslinking curve shows that EPDM with lower Mooney viscosity had a higher crosslinking density. For TPV, the nucleation of the crosslinking point for EPDM with lower Mooney viscosity improves the crystallization ability of PP, which is better than 4770P TPV, as evidenced by the higher crystallization temperature.

Ethylene in EPDM with higher Mooney viscosity is crystallized at room temperature, but the crystallization of ethylene in EPDM with lower Mooney viscosity cannot be obtained from the WAXD study at room temperature. The TPV with low Mooney viscosity EPDM has a two-phase structure, but the TPV with high EPDM Mooney viscosity has the better compatibility of PP and EPDM from the SAXS study and DMA analysis.

The higher crystallinity of PP in TPV with higher Mooney viscosity and the better compatibility between PP and EPDM lead to higher strength and elongation at break. The crystals in the EPDM phase can play a reinforcing role, leading to the improved mechanical property.

## Figures and Tables

**Figure 1 polymers-15-00642-f001:**
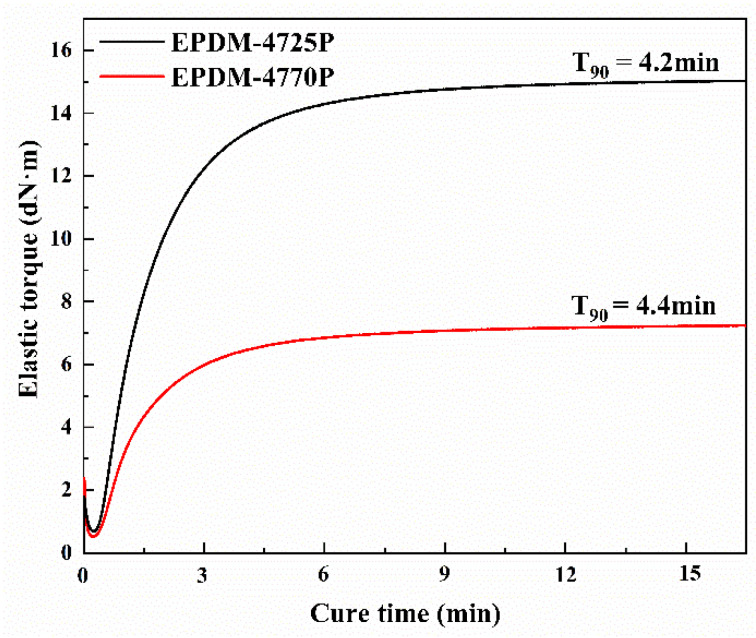
Crosslinking curve of EPDMs at 180 °C.

**Figure 2 polymers-15-00642-f002:**
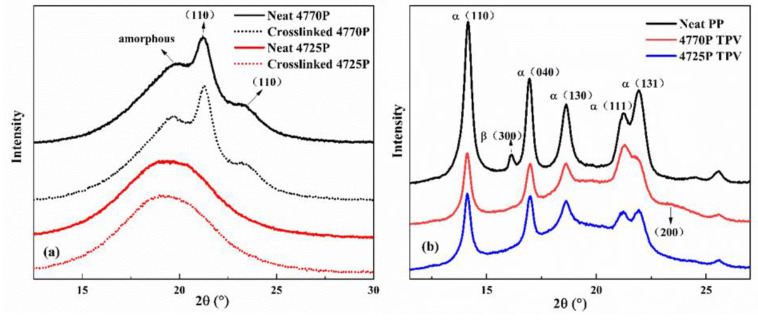
Wide-angle X-ray diffraction profiles for neat PP, neat EPDMs, crosslinked EPDMs, and TPVs: (**a**) EPDMs and crosslinked EPDMs; (**b**) neat PP, EPDM/PP thermoplastic vulcanizates.

**Figure 3 polymers-15-00642-f003:**
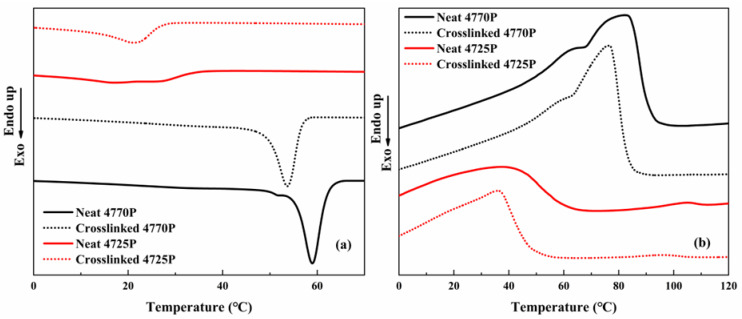
DSC cooling (10 °C/min) thermograms and the subsequent melting (10 °C/min) thermograms of neat 4770P, neat 4725P, crosslinked 4725P, and crosslinked 4770P: (**a**) cooling thermograms; (**b**) the subsequent melting thermograms.

**Figure 4 polymers-15-00642-f004:**
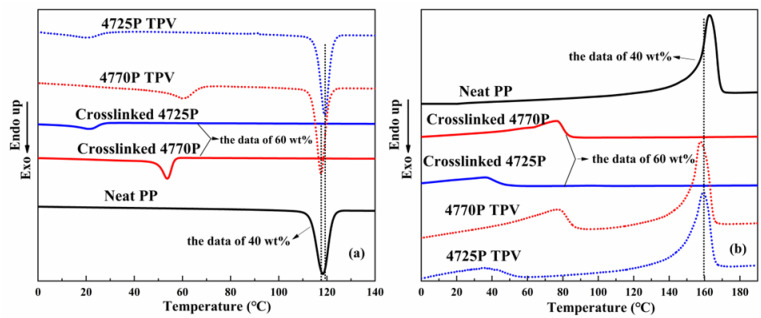
DSC cooling (10 °C/min) thermograms and the subsequent melting (10 °C/min) thermograms of neat PP, crosslinked EPDMs, and TPVs: (**a**) cooling thermograms; (**b**) the subsequent melting thermograms.

**Figure 5 polymers-15-00642-f005:**
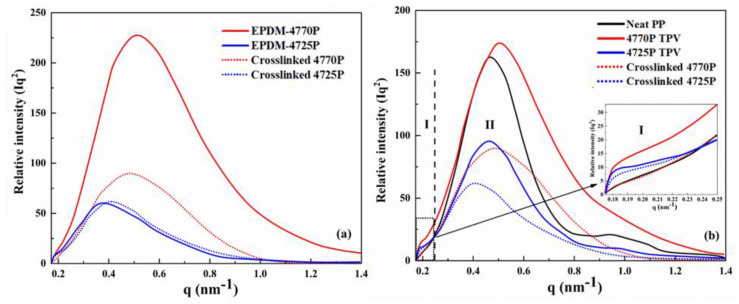
SAXS profiles of the samples: (**a**) The SAXS profiles of EPDMs and crosslinked EPDMs; (**b**) The SAXS profiles of neat PP, EPDM/PP thermoplastic vulcanizates, and crosslinked EPDMs: zone Ⅰ: q ≤ 0.25; zone Ⅱ: q > 0.25.

**Figure 6 polymers-15-00642-f006:**
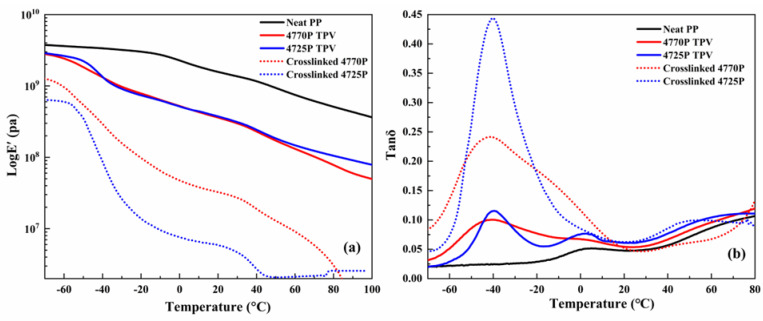
Storage modulus (**a**) and tan δ (**b**) of neat PP, TPVs, and crosslinked EPDMs.

**Figure 7 polymers-15-00642-f007:**
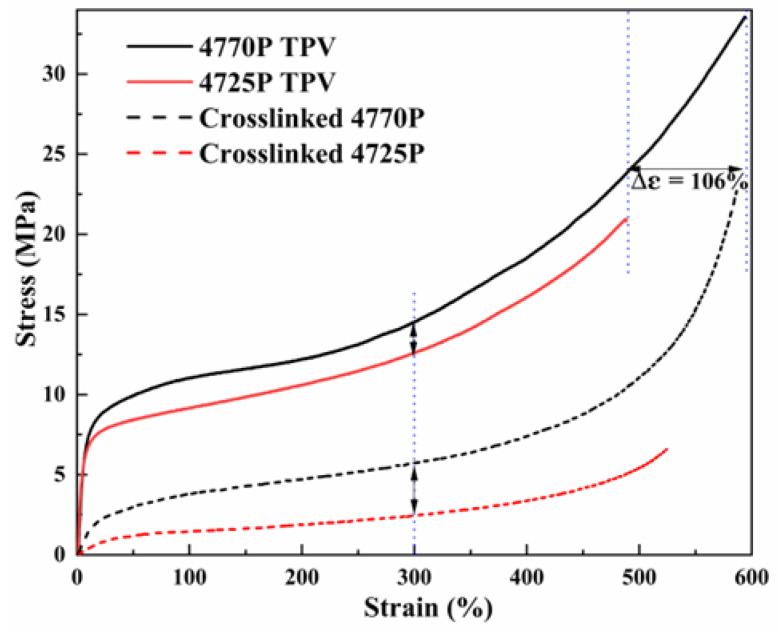
The stress-strain curves (up to break) of the crosslinked EPDMs and TPVs.

**Table 1 polymers-15-00642-t001:** The physical information of EPDMs.

EPDM	Mooney Viscosity (ML_1+4_, 125 °C)) (MU)	Ethylene Content (wt%)	Propylene Content (wt%)	Ethylidene Norbornene (ENB) Content (wt%)
NORDEL™ IP 4770P	70	70.0	25.0	5.0
NORDEL™ IP 4725P	25	70.0	25.0	5.0

**Table 2 polymers-15-00642-t002:** The composition proportion of neat PP, crosslinked EPDMs, and EPDM/PP thermoplastic vulcanizates.

Samples	Relative Amount by Weight (wt%)				
	EPDM4770P	EPDM4725P	PP	DCP	Antioxidant 1010
Neat PP	0	0	100	0	0
Crosslinked 4770P	100	0	0	2	0
Crosslinked 4725P	0	100	0	2	0
4770P TPV	60	0	40	1.2	1
4725P TPV	0	60	40	1.2	1

**Table 3 polymers-15-00642-t003:** Crystallization temperatures, enthalpies of fusion, crystallinities, and melting points of neat PP and crosslinked EPDMs.

Sample	Tc (°C)	△H_f_ (J/g)	Tm (°C)	Crystallinity of Components (wt%)
Neat PP	118	98.1	165	46.9
Crosslinked 4770P	54	66.6	77	22.7
Crosslinked 4725P	21	37.7	36	12.7

**Table 4 polymers-15-00642-t004:** Crystallization temperatures, enthalpy of fusion, crystallinities, and melting points of 4770 TPV and 4725 TPV.

Sample	T_C_		△Hf		Tm		Crystallinity of Components (wt%)	
	Tc_1_ (°C)	Tc_2_ (°C)	△Hf_1_ (J/g)	△Hf_2_ (J/g)	Tm_1_ (°C)	Tm_2_ (°C)	Crystallinity of EPDM (wt%)	Crystallinity of PP (wt%)
4770P TPV	60	118	26.2	40.8	77	158	14.9	48.8
4725P TPV	21	119	19.4	39.7	36	159	11.0	47.5

**Table 5 polymers-15-00642-t005:** Mechanical property data of the crosslinked EPDMs and TPVs.

Sample	Tensile Strength (MPa)	Elongation at Break (%)	Tensile Stress at 100% (MPa)	Tensile Stress at 300% (MPa)	Permanent Tensile Deformation (%)
Crosslinked 4770P	22.7	587.1	3.7	5.0	250
Crosslinked 4725P	6.6	524.2	1.5	2.5	65
4770P TPV	33.6	594.2	11.0	14.5	375
4725P TPV	20.9	488.3	9.2	12.6	300

## Data Availability

All the data are available within this manuscript.

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
