# Peer review of "Effects of Ethylene-Propylene-Diene Monomers (EPDMs) with Different Moony Viscosity on Crystallization Behavior, Structure, and Mechanical Properties of Thermoplastic Vulcanizates (TPVs)"

_polymers, 2023, doi:10.3390/polym15030642_

Round 1

Reviewer 1 Report

Comments:

1.     It is not proper to use abbreviations of EPDM or TPV in article title.

2.     Fig. 1. What is the meaning of “Mooney viscosity of 70”? Is it understandable to readers? In addition, Fig. 1 is torque vs. time. How and why the torque is related to “crosslinking curves”? 

3.     Fig. 2 WAXD. You have two graphs in Fig. 2, but only one figure caption. Which one is what? Discussion has “Figure 2(a)”, but there are no labels for these graphs.

4.     Same confusion in Fig. 3 DSC, with two graphs and only figure caption, which does not clearly specify which one is what? In addition, in the y-axis, “Endo up” is labeled, but there is a downward arrow indicating “heat flow”.  This is confusing to readers. Are these labels clear to readers?

5.     Writing discrepancy in p. 6. The sentence of “The effect of crosslinked EPDMs on the crystallization behavior of the TPV are presented in Figure 4.” Is placed at a paragraph detached from the discussion texts in another paragraph.

6.     Same confusion in Fig. 5 (SAXS), with two graphs and only figure caption does not clearly specify which one is what? and perhaps errors too. Clear labels are missing to tell which graph is for which system in figure caption. 

7.     Fig. 5. In addition, why one is q and the other is q2 in x-axis, but both have same unit?

8.     Fig. 5 and Fig. 6. Synchrotron SAXS data. What is the exact physical meaning of “long period” in this work? The “long period” is used to represent the averaged inter-distance of neighboring crystals in the sample.  This question is raised, because obviously, in this work, PP is semicrystalline, and it has lamellae and spherulites; however, the other segments may not have lamellae or crystals. So what do they mean that the “long periods” from 12-14 nm?  Are these values referring to lamellae or other domains?

9.     The system in this work contains blends of PP and EPDM to form TPV. There are PP crystals and phase domains of blends. So in Fig. 5 and Fig. 6, how can the parameters of “long periods” from 12-14 nm be assigned to any specific orders in blend of PP and EPDM?

10.  Fig. 7. DMA. The data of DMA are shown in Fig. 7a,b (E’ and tan-delta). Authors discussed: “ It can be seen that neat PP shows two relaxation peaks at temperature 82 °C (α relaxation) and around 0 °C (β relaxation), respectively.”, etc…..  This referee wonders and wants to ask what are the meanings of these two assignments of molecular relaxation? PP has a Tg at -20~-25oC. Why it will show two relaxation peals at 82 °C (α relaxation) and 0 °C (β relaxation) in the DMA data of this work?  Can authors justify the interpretation?  Are there any relevant literature backgrounds to support PP has 82 °C (α relaxation) and 0 °C (β relaxation)? If not, were the DMA characterization and data acquisition properly done?

11.  Fig. 7, p. 10. The assignments of numerous alpha-relaxation or beta-relaxation peaks for crosslinked EPDMs and TPVs, etc. are quite confusing, erratic and speculative without solid grounds.  

12.  By judging from the E’ data for crosslinked 4725P, between T= 20 to 100oC, portions of experiments appeared to be improperly carried out, leading to erroneous interpretation of tan(delta) peaks in discussion texts of “4725P TPV exhibits three peaks: a sharp β relaxation peak at -39.7 °C which corresponds to the 4725P rubber phase, a peak at 3 °C (second peak) and a weak peak at 70 °C (third peak) relate to the β transition of PP phase and α transition of PP phase, respectively.” 

Authors are better to re-characterize DMA of the specimens for accuracy.

Reviewer 2 Report

In the paper, an effect of Mooney viscosity of Ethylene-Propylene-Diene Monomer (EPDMs) on the crystallization dynamics, crystal structure and properties of the EPDM/Polypropylene (PP) based thermoplastic vulcanizates (TPVs) has been studied.

The paper is interesting, valuable, and quite well written, however, mostly of an engineering value. This is an engineering project, and the authors has solved an engineering problem. They should expand the scientific elements of their studies.

The authors should indicate the novelty elements of their study and try to explain the obtained results. Only such an explanation determines the scientific nature of the research performed.

The authors comment the result of experimentation in details. However, there is lack of short and clear statements on the novelty of this research. The authors should try to generalize the results of their research, not just report them.

Moreover, the material and sample preparation should be better described. For example, extrusion should be better reported, screw geometry should be presented which is crucial for sample preparation.

There are also minor errors, for example on p.4 “The crosslinking curves of the EPDM-4770P (Mooney viscosity of 70) and EPDM-4725P (Mooney viscosity of 70) are illustrated in Figure 1”.

Reviewer 3 Report

After review the manuscript, there are several points that need to be corrected/clarified, following they are detailed:

- Please check, that first time an abbreviation is written, this must be defined, for instance DCP in page 1.

- For table 2, how decided that EPDM proportion must be 60wt%? and also why for TPV DCP content decrease from 2wt% to 1.2wt%?

- For XRD results discussion, please cited similar works that indicate the presence os beta peaks of PP is good in TPV.

- For some figures (2, 3, 5) please indicate in caption, for a and b images which materials are reported.

- For temperatures reported from thermal analysis, please delete decimals after the point, due they are not significant. Also, in some cases indicate the there is a temperature increasing, when the variation is 1-2ºC, in thermal analysis it is necessary that temperature shows a variation of at least 5ºC for consider that there is a change in property, so in this case there are not significant variations in temperatures.

- For table 4 caption please correct, due it is not clear enough indicating that composition proportion is reported, but table data correspond to thermal properties obtained from DSC analysis, so the caption must corresponds to information reported.

- If neat PP was included in fire 6, long period, why did not include EPDM 4770P and 4725P samples?

- Bibliography indicate that molecular weight causes a longer plateau region in Storage modulus curves, in this case this is not occurs, why? The change is present only at higher temperature than 70ºC for TPVs. Also, how crystallinity is affected for molecular weight?

- DMA is reported as a good technique for calculation of crosslinking density, this is related to proportionally to the rubbery modulus. May be this could be calculated and compared with behavior observed in figure 1.

- The referred weaker alpha relaxation transition  radon -2ºC (page 10) for crosslinked 4770P, this is not visible in Tan delta curve, so please clarify this or indicate it with an arrow in figure. From my point of view it seems more the broad form of peak than a weaker relaxation.

Round 2

Reviewer 1 Report

Revised version is acceptable, including DMA re-characterization.

Reviewer 3 Report

After review the corrected version, I wish to thanks to authors for consider my previous recommendation/observations, in the aim to improve the manuscript which not shows a significant positive change compared with first version, so based on this I can recommend the Acceptation of manuscript.